# Comparison of Physicochemical Properties of Noodles Fortified with Commercial Calcium Salts versus Calcium Citrate from Oyster Shells

**DOI:** 10.3390/foods12142696

**Published:** 2023-07-13

**Authors:** Hong-Ting Victor Lin, Guan-Wen Chen, Ke-Liang Bruce Chang, Yi-Jun Bo, Wen-Chieh Sung

**Affiliations:** 1Department of Food Science, National Taiwan Ocean University, 2 Pei-Ning Road, Keelung 20224, Taiwan; hl358@mail.ntou.edu.tw (H.-T.V.L.); klchang@mail.ntou.edu.tw (K.-L.B.C.); 10032057@mail.ntou.edu.tw (Y.-J.B.); 2Center of Excellence for the Oceans, National Taiwan Ocean University, 2 Pei-Ning Road, Keelung 20224, Taiwan

**Keywords:** calcium citrate, calcium fortification, noodle quality, oyster shells

## Abstract

This study examined the physicochemical effects of the fortification of noodles with 0.25–1.00% (*w*/*w*) calcium salts, viz. calcium acetate, calcium carbonate, calcium citrate, and calcium lactate. Fortification with calcium citrate, calcium acetate, and calcium carbonate increased the pH and breaking force of the dried noodles. However, the fortification of noodles with any concentration of calcium did not increase the extent of elongation of the control raw noodles. The L* and b* values of the raw and dried noodle color increased with increasing concentrations of calcium salts, except for noodles with added calcium citrate. Fortification with calcium citrate yielded no significant influence on color, texture, adhesiveness, springiness, flavor, and overall scores for cooked noodles. Noodles fortified with 0.5% calcium citrate made from oyster shells were compared with a control sample of noodles and noodles fortified with commercially available calcium citrate. The particle size of the calcium citrate made from oyster shells (258 nm) was smaller than that of the purchased calcium citrate (2631 nm). Noodles fortified with calcium citrate made from oyster shells showed no significantly difference compared to noodles fortified with commercially available calcium citrate. These results suggest that calcium citrate made from oyster shells may be used as the additive of choice for the manufacture of calcium-fortified noodles.

## 1. Introduction

Noodles are a staple food in many Asian countries and account for almost 40% of all wheat products consumed in Asia [1]. Yellow alkaline noodles and white salted noodles are two widely consumed types of noodles. White salted noodles are made from all-purpose flour, water, and salt. When assessing the quality of noodles, Asian consumers consider the primary factors of color and texture [1]. Asian consumers prefer white salted noodles made from refined all-purpose flour that is light and has a low ash content. White salted noodles lack dietary fiber, minerals, vitamins, and various essential amino acids, such as lysine [2]. Noodles fortified with ground vegetables or eggs have been developed. Fortified noodles have become popular in Taiwan in recent years. Calcium salt-fortified rice noodles showed no significant changes in cooking loss, cooking yields, sensory acceptability, and texture, although it took 10 min more at 90 °C to cook than control rice noodles [3]. 

Calcium lysinate and calcium carbonate showed greater bioaccessibility than tricalcium phosphate in cooked rice noodles. Whether wheat noodles can be fortified with calcium has not been explored. Depending on the ingredients, fortification of noodles could weaken the gluten network causing the loss of solids during cooking.

In addition to its nutritional value, calcium plays an important role as an intracellular messenger [4]. Calcium intake deficiency leads to the development of osteoporosis and it is a main public health concern issue. Calcium is essential for the growth of strong bones and teeth. Calcium that circulates in the blood is involved in blood clotting and muscle contraction by helping to regulate blood clotting and muscle contraction and also helps to regulate blood pressure [5]. In the past few years, interest in the nutritional value of food has increased among consumers, with a corresponding increase in the demand for functional foods fortified with calcium. The recommended dietary allowances (RDA) for calcium are 600 mg for adults and 1–9-year-old children [6]. Calcium supplements made from oyster shells and limestone have been developed. Lee et al. discovered that nanocalcium particles made from oyster shells effectively prevented osteoporosis among rats [7]. According to Greger et al., the relatively high bioavailability of calcium in products made from oyster shells (102%) or in those containing calcium lactate (104%), calcium carbonate (102%), and calcium citrate (85–95%), make these products excellent sources of calcium compared with dried skim milk, which has a bioavailability of 100% [8]. Roland studied the particle size and solubility of calcium from egg and oyster shells, concluding that oyster shells were a better source of calcium than were egg shells [9]. The L-aspartic acid chelated calcium from oyster shell could improve calcium absorption rate and increase the serum biochemical index, biomechanical properties, and bone growth indexes [10]. Calcium carbonate and phosphate are ideal for food fortification due to their lower solubility and low cost in foods and they were recognized as safe by the USDA [11]. 

Sodium carbonate and potassium carbonate (0.3–1.0%) are the most common alkaline salts (kansui) added to fresh yellow alkaline noodles [12,13]. These salts increase the dough development time and water absorption to produce a slightly extendable and firmer noodle dough and giving it a unique yellow color [14]. 

A total of 220,000 tons of oyster shells are produced each year at oyster processing plants that process oysters harvested from the western coastal regions of Taiwan [15]. The proper disposal of waste oyster shells has become a challenging issue in the oyster aquaculture industry. Food products fortified with calcium from oyster shells have been demonstrated to prevent and treat osteoporosis among rats and humans [16,17].

This study evaluated the effects of four calcium salts including calcium acetate, calcium carbonate, calcium citrate, and calcium lactate on the appearance, cooked quality, taste, and physicochemical properties of raw and dried white salted noodles. Calcium citrate is a proven food supplement but the calcium citrate made from oyster shells has not been evaluated in fortified noodles. The optimal formulation including calcium salts made from oyster shells was determined, and the sensory properties of fortified white salted noodles were evaluated to provide scientific basis for developing calcium supplemented noodles and utilizing oyster shells.

## 2. Materials and Methods

### 2.1. Raw Materials and Chemicals

All-purpose flour was obtained from Uni-President Corporation (Tainan, Taiwan). Sodium hypochlorite was purchased from Hong Sheng Instruments (Taipei, Taiwan). Acetic acid and citric acid were purchased from J.T. Baker (Phillipsburg, NJ, USA). Calcium salts (extra-pure reagents of calcium lactate, calcium acetate, calcium carbonate, and calcium citrate) were obtained from Ninhon Shiyaku Industries (Taipei, Taiwan). Oyster shells were obtained from the Dongshih oyster processing plant (Dongshih Township, Chiayi County, Taiwan).

### 2.2. Preparation of Calcium Citrate from Oyster Shells

Oyster shells were washed with tap water and then brushed to remove any remaining materials from the surface. The shells were soaked in a 6% sodium hypochlorite and water solution, the volume of which was 10 times that of the shells, and the shells were then sonicated for 30 min in an ultrasonic cleaning machine (5510R-DT21, Branson, MI, USA). The shells were washed with tap water and dried at 45 °C overnight. The dried shells were crushed with a pulverizer (D3V-10, Yu Chi Machinery, Chang hua, Taiwan) for 1 min. The homogenized shell powder was passed through a mesh (No. 120) and stored in a closed container. The shell powder was dissolved in 1M acetic acid, the volume of which was 40 times that of the shell powder, and the powder was heated to 90 °C following the conditions of Wang et al. [18]. The shell powder was then filtered. Citric acid was added to the dissolved calcium acetate solution and allowed to settle for 12 h. The precipitated calcium citrate was filtered, washed twice with deionized water, oven dried at 50 °C, and stored at room temperature awaiting further analysis.

### 2.3. Fourier Transform Infrared Spectroscopy and Evaluation of Calcium Salt Particle Size Distribution

Fourier transform infrared spectra of the calcium salt samples were analyzed according to the method of Hofmann et al. [19] at 25 ± 2 °C. Calcium citrate and potassium bromate (1:100) were mixed and then oven dried at 60 °C in an oven for 24 h. The mixture was loaded onto a crystal cell and clamped onto the mount of a Fourier transform infrared spectrometer (Spectrum Two; Perkin Elmer, Waltham, MA, USA). The transmission percentage was recorded in the 600–4000 cm^−1^ spectra range. 

Samples of calcium citrate from the oyster shells and the purchased calcium salts (0.1 g) were dispersed in double-distilled water and sonicated for 30 s. Each sample (700 μL) was loaded into a disposable capillary cell (DTS 1061, Malvern Instruments, Worcestershire, UK). The hydrodynamic diameter was determined using a dynamic light scattering analyzer (Malvern System 4700c Sub-Micron Particle Analyzer; Malvern Instruments, Worcestershire, UK). 

### 2.4. Noodle Preparation and Physicochemical Properties of Noodles

The method for preparing the white salted noodles was adapted from Li et al. [20]. White salted noodles were made from all-purpose flour (100.0 g), distilled water (45.0 g), salt (2.0 g), and various concentrations (0.25%, 0.50%, 0.75%, and 1.00% elemental calcium *w/w*) of calcium salts (calcium lactate, calcium citrate, calcium acetate, and calcium carbonate) or 0.50% calcium citrate from oyster shells. A sample of white salted noodles was also prepared without calcium fortification. The sample mixtures were kneaded to form crumbly dough balls and allowed to rest for 30 min. The dough balls were rolled and folded in half on a Marcato Atlas 150 Wellness Pasta Maker (Campodarsego, Italy) to form sheets. The dough sheets were passed through the pasta maker twice, three times at progressively decreasing gaps of 2.50, 1.85, and 1.45 mm, respectively. The dough sheets were then cut into strips of about 15–20 cm in length and 0.65 cm in width, half of which were left raw and half of which were oven dried at 45 °C for 5 h to obtain dried noodles having a moisture content of less than 15% and water activity of less than 0.65. Noodle manufacturers generally use temperatures lower than 45 °C to operate the drying process [21].

The effect of the calcium salts on the pH of the noodles was measured according to the method of Hwang, Sung, and Shyu [22]. Moisture and ash contents were measured according to the AOAC method 935.29 and method 923.03, respectively [23]. Water activity was measured using a Novasina Thermoconstanter (RTD 33 TH-1; Novasina, Pfaffikon, Switzerland) following the method of Koh et al. [24]. Cooking yield and loss were determined according to the AACC method 66–50 [25].

### 2.5. Determination of Calcium Content, Color, and Texture of Noodles

The calcium content of the dried noodles was determined according to the method of Choo and Aziz [26], which involves using an atomic absorption spectrophotometer (5100PC; Perkin Elmer, Norwalk, CT, USA). Measurements were made at a wavelength of 422.7 nm. Absorbancies were recorded and a standard curve of calcium ion solution was plotted. Results were expressed as mg/L of the sample. Color was examined on a spectrocolorimeter (TC-1800 MK II, Tokyo, Japan) using the L* (lightness), a* (redness/greenness), and b* (yellowness/blueness) color scales following the method of Good [27] and Janve and Singhal [3] with some modification. A white tile and a black cup were used to standardize the spectrocolorimeter. Color was recorded by taking six measurements for each sample, and triplicate determinations were made per measurement. Color difference (ΔE) and white index (WI) were obtained using the following formulas: ΔE = [(ΔL*)^2^ + (Δa*)^2^ + (Δb*)^2^]^1/2^(1)
ΔL* = L*_sample_ − L*_control_(2)
Δa* = a*_sample_ − a* _control_(3)
Δb* = b* _sample_ − b* _control_(4)
WI = 100 − [(100 − L*)^2^ + (a*)^2^ + (b*)^2^]^1/2^(5)
where the L*, a*, and b* color scales correspond to the CIE color parameters of the noodles. 

Tensile strength of the dried and raw noodles was determined using a method adapted from Reungmaneepaitoon, Sikkhamondhol, and Tiangpook [28]. A Code A/CKB probe was used with the following operating parameters: trigger point, 3 g; pre-test speed, 2.0 mm s^−1^, test and post-test speed, 1.5 mms^−1^; and deformation, 90%, distance, 30 mm for the breaking force, breaking point (mm), and work of breaking (N × mm) of dried noodles. Tensile strength was measured using an A/SPR probe attached to a texture analyzer (TA-TX2; Stable Micro Systems, Godalming, UK) with a 5 kg load cell. Noodles were positioned on the rig arm slots of the texture analyzer, and the load cell was raised until the noodle strands ruptured. Rig calibration with a return trigger path at 30 mm was performed before the analysis. The rig speed was set to 1.5 mm/s. Tensile force was expressed as the maximum force (N) of the rig required to pull raw noodles 250 mm apart. Tensile elongation was expressed as the length (mm) of the noodle strand rupture. 

### 2.6. Sensory Evaluation of Noodle Fortified with Calcium Citrate

Cooked, calcium-fortified noodles were served to 65 untrained panelists recruited from the Department of Food Science, who evaluated their appearance, texture, adhesiveness, springiness, and flavor. Attributes were rated on a 7-point hedonic scale with endpoints ranging from 1 = (extremely dislike) to 7 = (extremely like). Noodle samples were coded with three-digit random numbers.

### 2.7. Statistical Analysis

All analyses were carried out in three replications per treatment unless described otherwise. Analysis of variance was performed using SPSS (version 1.2, 1998). Duncan’s multiple range test was used to identify differences between treatments at a 5% significance level (*p* < 0.05).

## 3. Results and Discussion

### 3.1. Particle Sizes of Calcium Salts

The particle sizes of the calcium lactate, citrate, acetate, and carbonate salts were 451, 2631, 886, and 1877 nm, respectively. The particle size of the calcium citrate made from oyster shells was 258 nm. In another study, calcium carbonate nanoparticles had a particle size of 100 nm (z-average = 100 nm) [29]. Heller et al. (1999) [30] reported that the bioavailability of calcium citrate was higher than that of calcium carbonate. However, the small particle size of calcium citrate made from oyster shells may dissolve in water much easier and convert to a calcium ion, but it may not have higher bioavailability than those of commercially available calcium citrate salt.

### 3.2. Fourier Transform Infrared Spectroscopy of Calcium Salts

Typically, the main component of oyster shells is calcium carbonate, with absorption peaks at 874 cm^−1^ and 1408 cm^−1^. Shan et al. reported similar spectra for calcium carbonate nanoparticles [31]. Absorption peaks appeared at 841, 1086, 1433, and 1545 cm^−1^ for the calcium citrate made from oyster shells (Figure 1A). Absorption peaks appeared at 837, 1078, 1435, and 1539 cm^−1^ for both the purchased calcium citrate and pure (>98%) calcium citrate (Figure 1B,C). The absorption peak ranges of the purchased calcium citrate and calcium citrate made from oyster shells were similar, indicating that the calcium carbonate content of the oyster shells (Appendix A) had been successfully converted into calcium citrate.

### 3.3. Moisture Content and Water Activity of White Salted Noodles

The moisture contents of raw noodles with 0%, 0.25%, 0.50%, 0.75%, and 1.00% calcium citrate were 35.9%, 35.5%, 35.5%, 34.9%, and 33.8% water, respectively (Figure 2A). The difference in the moisture content between the 0.75% or 1.00% calcium citrate-fortified raw noodles and the control raw noodles was significant (*p* < 0.05). The water activities of the control and calcium citrate-fortified raw noodles were 0.943 and 0.941, respectively. No change in water activity was observed when an increasing concentration of calcium citrate was used to fortify the raw noodles, indicating that moisture content does not correspond to water activity.

By contrast, the concentration of calcium acetate was significantly inversely correlated with water activity (*p* < 0.05). The water activities of 0.25%, 0.50%, 0.75%, and 1.00% calcium acetate-fortified raw noodles were 0.931, 0.923, 0.911, and 0.900, respectively. Even at lower dose (0.25%), relative to the control sample, a notable decrease in water activity was observed. Because of this property, calcium acetate is a more promising option than the other calcium salts for fortification purposes. 

The moisture contents of the control dried noodles and 1% calcium citrate-fortified dried noodles (oven dried at 105 °C according to the AOAC method) were 9.6% and 8.5%, respectively. The moisture content of dried noodles fortified with calcium acetate or calcium lactate was higher than that of the control dried noodles, possibly because calcium acetate and lactate tend to more effectively bind with water molecules than do calcium carbonate or citrate (Figure 2B). 

Fortification with calcium citrate or lactate at concentrations higher than 0.75% significantly decreased water activity in dried noodles (*p* < 0.05). The water activities of the control, 0.75% calcium citrate-fortified, and 1.00% calcium citrate-fortified dried noodles were 0.529, 0.513, and 0.463, respectively. The water activity of the dried noodles substantially decreased at calcium lactate concentrations of 0.75% (water activity = 0.491) and 1.00% (water activity = 0.472). Calcium acetate and carbonate fortification did not effectively decrease water activity relative to the control dried noodles.

Although the moisture contents of calcium carbonate-fortified and calcium citrate-fortified dried noodles were lower than that of control dried noodles, fortification with 1.00% calcium citrate yielded the greatest reduction in water activity (Figure 2B). The water activities of the control, ≥0.5% calcium citrate-fortified, and ≥0.5% calcium lactate-fortified dried noodles were low enough that these samples could be considered safe to eat.

### 3.4. Ash and Calcium Content of White Salted Noodles

The crude ash content of the control raw noodles was 0.52%, and fortification with 0.25%, 0.50%, 0.75%, and 1.00% calcium acetate increased the crude ash contents by 42%, 85%, 100%, and 155%, respectively. Calcium salts significantly increased the crude ash content in the raw noodles. Fortification with other calcium salts also increased the ash content of the dried noodles. The effects of calcium salts on the ash content of both raw and dried noodles were similar. Atomic absorption spectrophotometer data show the control, 0.25%, 0.50%, 0.75%, and 1.99% calcium salt-fortified dried noodles contained 22 ppm, 218–239 ppm, 407–449 ppm, 611–638 ppm, and 792–891 ppm calcium (Appendix A). The maximum level of calcium daily dietary intake is not over 1500 mg calcium for children and adults. The recommended dietary allowances (RDA) for calcium are 600 mg for adults and 1–9-year-old children [6]. Therefore, it is not recommended that more than 6 kg of noodles per day of the 0.50% calcium citrate-fortified dried is eaten.

### 3.5. pH of Noodles

The pH values of the control and 1.00% calcium carbonate-fortified raw noodles were 5.90 and 7.73, respectively. Fortification with calcium lactate decreased the pH of the raw noodles to pH 5.5–6.0. By contrast, fortification with calcium citrate, calcium acetate, and calcium carbonate increased the pH of the raw noodles (Figure 3A). The pH values of the dried noodles fortified with various calcium salts exhibited similar patterns to those of the raw noodles fortified with various calcium salts (Figure 3B). Calcium ions were found to increase pH by competitively displacing protons from ionizable oxygen, nitrogen, or sulfur atoms that share electrons with hydrogen atoms. This mechanism occurs when the pH is sufficiently high to allow for mass action competition between protons and calcium ions [32].

The pH value of rice flour has been demonstrated to influence pasting and gelatinization characteristics. Viscosity decreased when rice flours were pasted at low pH values [33]. Using water with a pH of 5–6 to cook the noodles caused less cooking loss than using water with other pH values. The 0.75% and 1.00% calcium lactate-fortified noodles had the highest cooking loss (Figure 4).

### 3.6. Cooking Qualities of Noodle

The quality of cooked noodles is the most important factor influencing the noodle purchase decision-making process of consumers [20]. The cooking yields of most of the calcium salt-fortified raw noodles decreased after 8 min of cooking relative to the control raw noodles (Figure 5A). The cooking yield of the raw noodles decreased, and the cooking loss increased, especially for the calcium acetate-fortified noodles (Figure 4A). The weakening of the gluten network resulting from the dilution of the gluten fraction after the incorporation of calcium acetate resulted in a lower cooking yield after the noodles were cooked. Cooking losses were positively correlated with the concentrations of calcium salts (Figure 4A). This observation implies that the calcium salts weakened the gluten network in the raw noodles.

Although the dried noodles containing calcium citrate or calcium lactate exhibited significant increases in cooking yields (Figure 5B), only those containing calcium citrate exhibited a significant decrease in cooking losses (Figure 4B). This implies that increasing the concentration of calcium citrate in dried noodles enhances water retention and prevents cooking loss during cooking. 

Fortification of the raw noodles with calcium salts at concentrations of more than 0.5% may increase cooking loss. Fortification with 0.25% calcium carbonate or calcium citrate decreased cooking losses (Figure 4A). Calcium carbonate and calcium citrate fortification decreased cooking losses to a significantly greater degree for the dried noodles than for the raw noodles (Figure 4B), possibly because the drying process caused the noodles to become denser, which more effectively prevented cooking water from penetrating into the core of the noodles. By contrast, concentrations of calcium lactate greater than 0.25% were sufficient to increase cooking losses in both raw and dried noodles (Figure 4). 

### 3.7. Physicochemical Properties and Color of Calcium-Fortified Noodles

Fortification of dried noodles with different concentrations of various calcium salts had significant effects on noodle strength. Breaking force, breaking point, and breaking work substantially increased for all calcium salt-fortified dried noodles (Table 1). Strength was positively correlated with the calcium salt concentration of the dried noodles. This finding suggests that calcium salts can be used with other ingredients to improve colors, flavors, and nutritional benefits while retaining the strength of noodles. The raw noodles fortified with 0.50% calcium citrates, 0.75% calcium acetate, or 1.00% calcium acetate exhibited the highest values of tensile strength and tensile elongation (Table 2). The raw noodles fortified with 0.50% calcium citrate exhibited the highest tensile strength. However, the noodles fortified with any concentration of calcium did not increase the extent of elongation of the control raw noodles, indicating that the gluten networks were not well formed in the calcium-fortified raw noodles. Fortification of the dried noodles with calcium citrate was more effective at increasing strength than was fortification with the other calcium salts (Table 1). The increase in dried noodle strength might be due to the pH change and the increased viscosity of noodle dough fortified with calcium citrate. The breaking forces of dried noodles fortified with calcium salts were higher than that of the control dried noodles, indicating that calcium addition to the dried raw noodles leads to increased binding strength. Calcium salts toughen noodle dough by interacting with gluten and extending dough development after the noodles have been dried.

### 3.8. Color of Noodles

Color is a critical factor influencing the consumer opinions of the quality of noodles [20]. The L* (67.86 to 77.07) and b* (31.75–34.70) values of the raw noodles increased with increasing concentrations of calcium salts, except for calcium citrate, and all raw noodles had a uniform light-yellow tinge (Table 3). Color measurements (the a* values of −8.21 to −9.63) showed a small variation in the a* values among the raw noodles. Change in the color of the raw noodles stored at 4 °C for 24 h was inappreciable. Nonsignificant changes in the white index of calcium-fortified raw noodles were detected for the raw noodles stored for 24 h. However, differences between the color values of the raw noodles and cooked noodles were observed. Higher L*, a*, and b* values, and white index values for the cooked noodles were observed relative to the control raw noodles. The color behaviors of the dried noodles fortified with different calcium salts were similar to those of the raw noodles fortified with the same calcium salts.

Relative to the control raw noodles, the color difference (ΔE) values for 1.00% calcium acetate, calcium carbonate, and calcium lactate-fortified noodles were all greater than three. These results indicate that the human eye can easily notice the difference between the control and 1% calcium salt-fortified raw noodles, except when calcium citrate was added (Table 3). The highest L* value and color difference (ΔE) values were observed for the noodles fortified with 1% calcium lactate when compared with the control raw noodles stored at 4 °C for one day. The dried noodles fortified with different concentrations of calcium citrate and lactate had significantly lighter surface color values than that of the control dried noodles. When 1% calcium salts were added to the noodle formulation, the difference in Commission internationale de l’eclairage (CIE) L*, a*, and b* values was significant, the difference between the calcium citrate and lactate-fortified noodles was obvious (Table 4). A noticeably lighter color was also observed for all 1.00% calcium salt-fortified dried noodles.

The dried noodles stored at room temperature (25 °C) for 24 h did not noticeably change color. However, differences between the color values of the dried noodles and cooked dried noodles were observed. Higher L*, a*, and b* values, and white index values were observed for the cooked dried noodles compared with the control dried noodles. The color values of L* and b* values of noodles decreased after drying. The white index decreased after the calcium salt-fortified noodles were dried.

Although calcium salts did not change in water activities in raw noodle but in dried noodles. Calcium citrate decreases the water content of raw and dried noodles, but it significantly increased the breaking force. Therefore, both water and calcium content influence physical and chemical properties of raw, dried, and cooked noodles in pH, cooking losses, cooking yield, breaking force, and L* values of dried noodles, and tensile force and elongation of raw noodles.

### 3.9. Sensory Evaluation of Fortified Noodles

No significant differences in color, texture, adhesiveness, springiness, flavor, and overall acceptability scores were observed between the cooked control noodles and noodles fortified with various concentrations of calcium citrate (Table 5). This finding indicates that the panelists could not tell the difference between the cooked noodles with 1.0% calcium citrate and the control noodles. Regarding the color and overall acceptability of the cooked noodles, the noodles fortified with 0.50% calcium citrate appeared to have a slightly higher color score than did the control noodles despite having similar overall acceptability scores. Calcium citrate was found to provide higher bioavailability than did calcium carbonate [26]. Fortification with calcium citrate did not induce an off-flavor or sticky taste. Although 1.00% calcium citrate fortification resulted in slightly lower texture, adhesiveness, springiness, flavor, and overall scores than those for the control noodles, the differences were not significant (*p* > 0.05). This suggests that the panelists accepted calcium citrate fortification at concentrations of up to 1.00%. Among the various concentrations of calcium citrate for fortification of the noodles, the concentration of 0.50% was found to be the most desirable in terms of overall acceptability. The noodles fortified with 0.5% calcium citrate made from oyster shells received comparable sensory scores. The sensory scores of the cooked dried noodles fortified with 0.50% calcium citrate made from oyster shells were slightly lower than those of the cooked control dried noodles. However, no significant differences were observed among the samples. These results suggest that fortifying white salted noodles with up to 0.50% calcium citrate retains cooking quality and improves dried noodle strength and appearance.

## 4. Conclusions

In summary, fortification with calcium salts altered the physicochemical properties favorably as seen from the increased calcium content of the raw and dried noodles and significantly increased the breaking force of dried noodles, which ensure the greater strength of dried salted noodles during packaging and transportation. Calcium salt addition to the white salted noodles led to a lighter color for the raw, dried, and cooked noodles. Of note, the cooking yields of the dried noodles fortified with more than 0.50% calcium citrate or calcium lactate were significantly higher than the other noodles. The calcium carbonate content of the oyster shells had been successfully converted into calcium citrate. Calcium citrate fortification (0.50%) had weaker negative effects on tensile strength and distance of raw noodles. The noodles fortified with 0.5% calcium citrate made from oyster shells received slightly higher sensory scores for the cooked dried noodles than did the noodles fortified with purchased calcium citrate. The positive results of this study may promote the further utilization of calcium recovered from waste oyster shells in noodle formulations. Calcium citrate made from oyster shells can be explored as well as calcium-fortified ingredients in yellow alkaline noodles for noodle manufacturing and enhancing the utilization of oyster shell resources to address the current calcium deficiency status in Asia. Additionally, it can also be evaluated for changes in physicochemical properties of yellow alkaline noodles.

## Figures and Tables

**Figure 1 foods-12-02696-f001:**
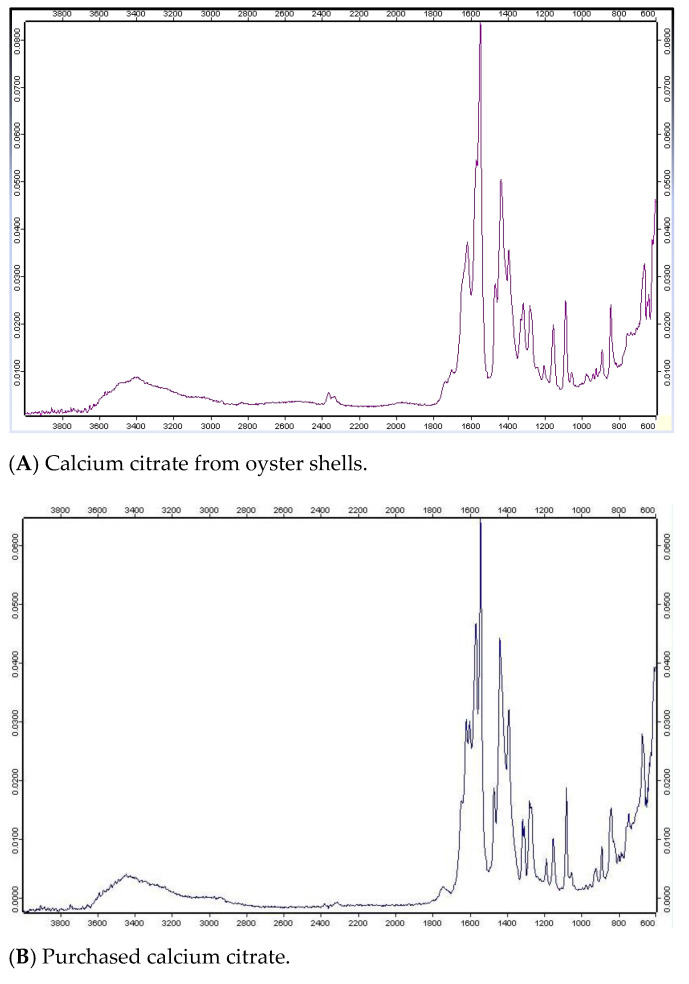
Infrared spectra of (**A**) calcium citrate from oyster shells, (**B**) purchased calcium citrate, and (**C**) pure (>98%) calcium citrate.

**Figure 2 foods-12-02696-f002:**
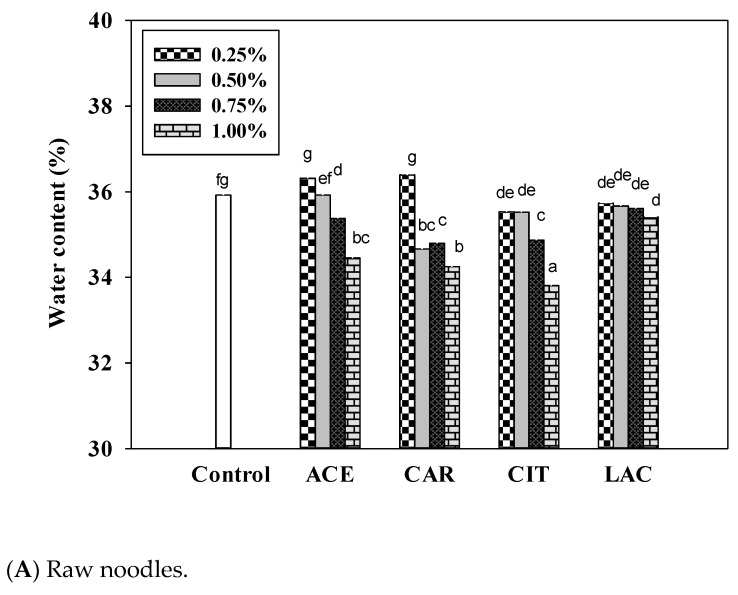
Effects of various concentrations of calcium salts on moisture content of (**A**) wet and (**B**) dried noodles. The superscripts indicate significant differences between different amounts of calcium salts. (*n* = 3; *p* < 0.05). ACE, calcium acetate; CAR, calcium carbonate; CIT, calcium citrate; LAC, calcium lactate.

**Figure 3 foods-12-02696-f003:**
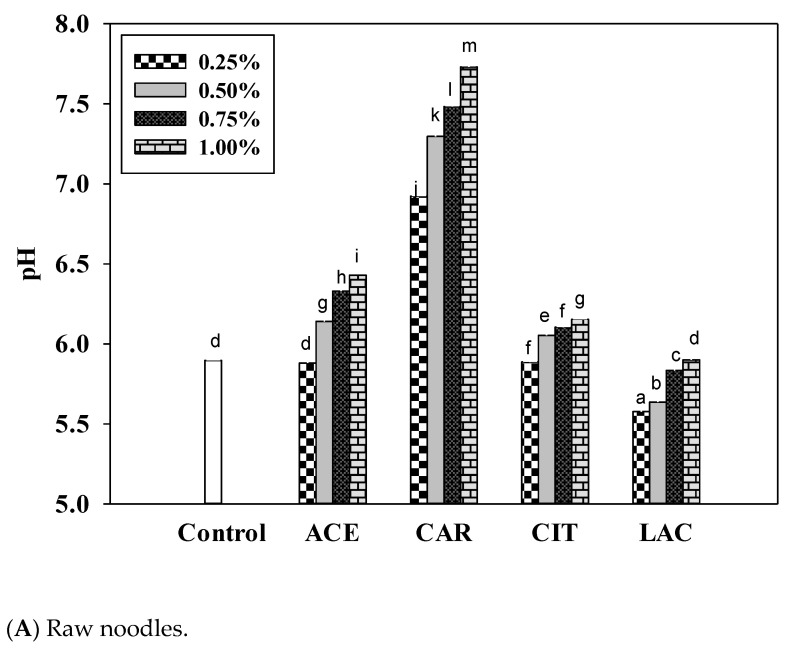
Effects of various concentrations of calcium salts on the pH of (**A**) wet and (**B**) dried noodles. Superscripts indicate significant differences between different concentrations of calcium salts. (*n* = 3; *p* < 0.05). ACE, calcium acetate; CAR, calcium carbonate; CIT, calcium citrate; LAC, calcium lactate.

**Figure 4 foods-12-02696-f004:**
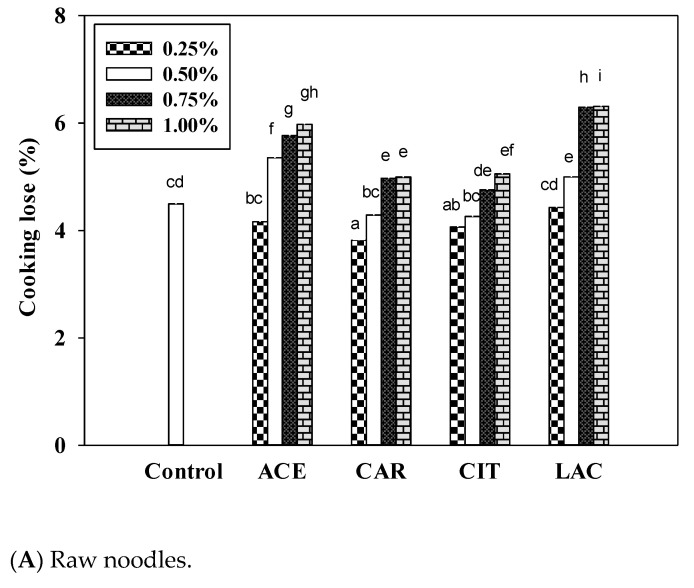
Effects of various concentrations of calcium salts on cooking losses of (**A**) wet and (**B**) dried noodles. Superscripts indicate significant differences between different concentrations of calcium salt. (*n* = 3; *p* < 0.05). ACE, calcium acetate; CAR, calcium carbonate; CIT, calcium citrate; LAC, calcium lactate.

**Figure 5 foods-12-02696-f005:**
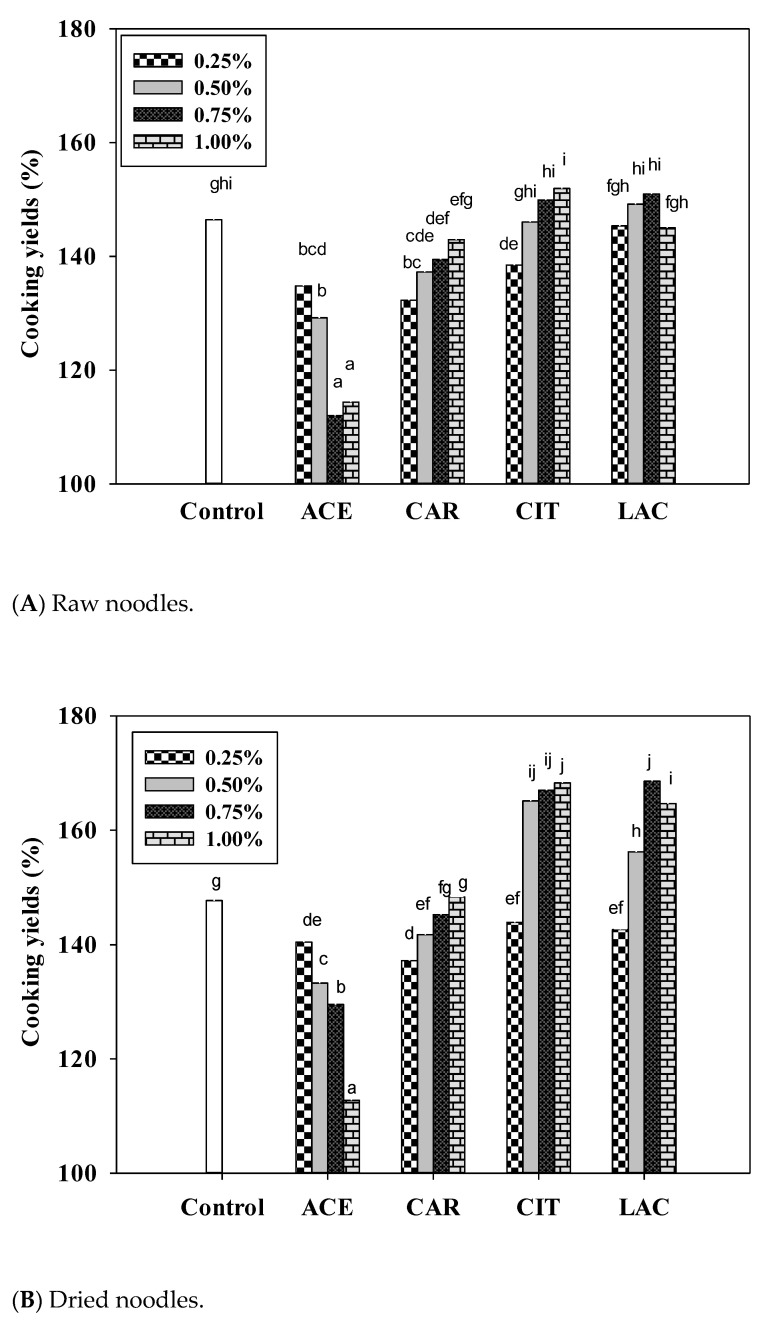
Effects of various concentrations of calcium salts on cooking yields of (**A**) wet and (**B**) dried noodles. Superscripts indicate significant differences between different concentrations of calcium salt. (*n* = 3; *p* < 0.05). ACE, calcium acetate; CAR, calcium carbonate; CIT, calcium citrate; LAC, calcium lactate.

**Table 1 foods-12-02696-t001:** Effects of different calcium salts on the textural characteristics of dried noodles.

Sample		Breaking Force * (N)	Breaking Point (mm)	Work of Breaking (N × mm)
Control		10.88 ± 0.47 ^a^	0.56 ± 0.03 ^a^	6.69 ± 0.09
ACE **	0.25%	18.39 ± 0.71 ^e^	0.60 ± 0.03 ^b^	11.08 ± 0.27
	0.50%	20.22 ± 0.83 ^h^	0.61 ± 0.02 ^b^	12.25 ± 0.91
	0.75%	22.91 ± 0.85 ^j^	0.78 ± 0.04 ^e^	17.75 ± 0.64
	1.00%	23.04 ± 0.53 ^j^	0.98 ± 0.05 ^h^	22.57 ± 1.30
CAR	0.25%	14.10 ± 0.61 ^b^	0.79 ± 0.04 ^e^	11.15 ± 0.83
	0.50%	15.81 ± 0.79 ^c^	0.85 ± 0.03 ^fg^	13.39 ± 1.15
	0.75%	17.62 ± 0.60 ^d^	0.83 ± 0.04 ^f^	14.59 ± 0.81
	1.00%	18.43 ± 0.92 ^e^	0.97 ± 0.05 ^h^	17.94 ± 1.66
CIT	0.25%	17.34 ± 0.78 ^d^	0.84 ± 0.03 ^fg^	14.59 ± 0.79
	0.50%	18.72 ± 0.36 ^ef^	0.87 ± 0.03 ^g^	16.24 ± 0.49
	0.75%	19.05 ± 0.21 ^ef^	0.95 ± 0.02 ^h^	18.05 ± 0.44
	1.00%	19.37 ± 0.85 ^fg^	0.97 ± 0.02 ^h^	18.85 ± 1.05
LAC	0.25%	21.49 ± 0.86 ^i^	0.68 ± 0.03 ^c^	14.67 ± 0.97
	0.50%	21.50 ± 0.91 ^i^	0.73 ± 0.03 ^d^	15.61 ± 1.37
	0.75%	19.73 ± 0.78 ^gh^	0.71 ± 0.03 ^cd^	13.99 ± 0.62
	1.00%	20.39 ± 0.95 ^h^	0.72 ± 0.03 ^d^	14.76 ± 0.55
Oyster CIT	0.50%	18.94 ± 0.25 ^c^	0.79 ± 0.04 ^b^	14.89 ± 0.75

* Mean (±standard deviation) values with different letters within the same column indicate significant difference at *p* < 0.05; *n* = 10. ** ACE, calcium acetate; CAR, calcium carbonate; CIT, calcium citrate; LAC, calcium lactate.

**Table 2 foods-12-02696-t002:** Effects of different calcium salts on tensile strength and tensile elongation of raw noodles.

Sample		Tensile Force * (N)	Tensile Elongation (mm)
Control		0.54 ± 0.06 ^gh^	19.08 ± 0.58 ^i^
ACE **	0.25%	0.46 ± 0.05 ^ef^	8.47 ± 0.30 ^b^
	0.50%	0.45 ± 0.03 ^ef^	12.39 ± 1.42 ^d^
	0.75%	0.53 ± 0.11 ^gh^	15.82 ± 0.34 ^f^
	1.00%	0.56 ± 0.08 ^ghi^	17.98 ± 0.43 ^h^
CAR	0.25%	0.40 ± 0.08 ^cde^	18.31 ± 0.71 ^h^
	0.50%	0.45 ± 0.03 ^ef^	11.51 ± 0.54 ^c^
	0.75%	0.39 ± 0.05 ^cde^	12.43 ± 0.26 ^d^
	1.00%	0.38 ± 0.04 ^bcde^	13.41 ± 0.37 ^e^
CIT	0.25%	0.42 ± 0.07 ^def^	13.53 ± 0.49 ^e^
	0.50%	0.58 ± 0.05 ^h^	16.85 ± 0.94 ^g^
	0.75%	0.50 ± 0.02 ^fg^	17.53 ± 0.70 ^gh^
	1.00%	0.50 ± 0.07 ^fg^	17.59 ± 1.21 ^gh^
LAC	0.25%	0.29 ± 0.04 ^a^	6.28 ± 0.41 ^a^
	0.50%	0.35 ± 0.03 ^abc^	6.44 ± 0.48 ^a^
	0.75%	0.35 ± 0.04 ^abcd^	5.77 ± 0.41 ^a^
	1.00%	0.31 ± 0.02 ^ab^	5.68 ± 0.52 ^a^
Oyster CIT	0.50%	0.60 ± 0.03 ^a^	17.98 ± 0.12 ^b^

* Mean (±standard deviation) values with different letters within the same column indicate significant difference at *p* < 0.05; *n* = 10. ** ACE, calcium acetate; CAR, calcium carbonate; CIT, calcium citrate; LAC, calcium lactate.

**Table 3 foods-12-02696-t003:** Effects of different calcium salts on the CIE L*, a*, and b* values * of raw noodles.

Sample		L*	a*	b*	ΔE	WI
Control		69.99 ± 0.06 ^b^	−8.21 ± 0.03 ^m^	32.15 ± 0.03 ^ef^	0.00	55.26
ACE **	0.25%	68.83 ± 0.02 ^a^	−8.32 ± 0.03 ^l^	31.94 ± 0.06 ^bcd^	1.20	54.58
	0.50%	70.07 ± 0.06 ^c^	−8.39 ± 0.01 ^k^	31.91 ± 0.10 ^bc^	0.27	55.46
	0.75%	72.06 ± 0.04 ^g^	−8.46 ± 0.01 ^j^	33.40 ± 0.02 ^k^	2.43	55.65
	1.00%	73.96 ± 0.07 ^fg^	−8.72 ± 0.02 ^f^	34.70 ± 0.03 ^n^	4.75	55.75
CAR	0.25%	68.58 ± 0.24 ^a^	−8.46 ± 0.01 ^j^	31.85 ± 0.03 ^ab^	1.46	54.47
	0.50%	72.42 ± 0.23 ^d^	−8.56 ± 0.01 ^h^	33.20 ± 0.20 ^j^	2.67	56.00
	0.75%	72.46 ± 0.08 ^e^	−8.67 ± 0.01 ^g^	33.32 ± 0.08 ^k^	2.77	55.92
	1.00%	74.49 ± 0.05 ^fg^	−9.09 ± 0.02 ^b^	34.23 ± 0.04 ^m^	5.04	56.36
CIT	0.25%	70.21 ± 0.05 ^fg^	−8.52 ± 0.02 ^i^	32.18 ± 0.03 ^fg^	0.38	55.32
	0.50%	70.15 ± 0.05 ^b^	−8.73 ± 0.02 ^f^	32.01 ± 0.06 ^cd^	0.60	55.34
	0.75%	71.96 ± 0.16 ^d^	−8.78 ± 0.01 ^e^	32.43 ± 0.05 ^h^	2.07	56.23
	1.00%	72.19 ± 0.20 ^e^	−8.81 ± 0.02 ^e^	32.30 ± 0.03 ^g^	2.27	56.49
LAC	0.25%	67.86 ± 0.03 ^d^	−8.37 ± 0.02 ^k^	31.75 ± 0.01 ^a^	2.18	54.05
	0.50%	70.59 ± 0.10 ^f^	−8.87 ± 0.03 ^d^	32.04 ± 0.03 ^de^	0.94	55.61
	0.75%	72.74 ± 0.16 ^h^	−8.92 ± 0.01 ^c^	32.69 ± 0.12 ^i^	2.88	56.52
	1.00%	77.07 ± 0.10 ^i^	−9.63 ± 0.02 ^a^	34.01 ± 0.03 ^l^	7.46	57.87

* Mean (±standard deviation) values with different letters within the same column indicate significant difference at *p* < 0.05; *n* = 3. ** ACE, calcium acetate; CAR, calcium carbonate; CIT, calcium citrate; LAC, calcium lactate.

**Table 4 foods-12-02696-t004:** Effects of different calcium salts on the CIE L*, a*, and b* values * of dried noodles.

Sample		L*	a*	b*	ΔE	WI
Control		61.55 ± 0.03 ^a^	−7.22 ± 0.01 ^b^	29.45 ± 0.01 ^d^	0.00	51.03
ACE **	0.25%	61.82 ± 0.08 ^b^	−7.13 ± 0.04 ^a^	29.50 ± 0.06 ^d^	0.29	51.22
	0.50%	62.75 ± 0.07 ^c^	−8.33 ± 0.02 ^f^	29.11 ± 0.03 ^c^	1.66	51.99
	0.75%	64.60 ± 0.04 ^e^	−8.33 ± 0.01 ^f^	30.12 ± 0.03 ^ef^	3.31	52.78
	1.00%	66.67 ± 0.14 ^f^	−9.60 ± 0.01 ^m^	30.75 ± 0.02 ^g^	5.79	53.65
CAR	0.25%	61.58 ± 0.02 ^ab^	−7.69 ± 0.02 ^d^	28.89 ± 0.03 ^b^	0.73	51.32
	0.50%	62.58 ± 0.09 ^c^	−8.14 ± 0.01 ^e^	28.03 ± 0.13 ^a^	1.98	52.54
	0.75%	64.75 ± 0.01 ^e^	−8.45 ± 0.05 ^g^	30.23 ± 0.11 ^f^	3.52	52.80
	1.00%	67.49 ± 0.31 ^g^	−9.22 ± 0.02 ^k^	31.24 ± 0.06 ^i^	6.52	53.98
CIT	0.25%	69.14 ± 0.21 ^i^	−8.64 ± 0.03 ^h^	31.00 ± 0.06 ^h^	7.88	55.41
	0.50%	70.19 ± 0.09 ^j^	−8.84 ± 0.02 ^i^	29.19 ± 0.02 ^c^	8.80	57.36
	0.75%	71.15 ± 0.25 ^k^	−9.15 ± 0.03 ^j^	31.42 ± 0.13 ^j^	11.71	56.37
	1.00%	72.66 ± 0.18 ^l^	−9.58 ± 0.09 ^m^	32.31 ± 0.16 ^k^	13.82	56.60
LAC	0.25%	69.27 ± 0.02 ^d^	−7.43 ± 0.01 ^c^	30.09 ± 0.02 ^e^	7.75	56.35
	0.50%	69.39 ± 0.03 ^h^	−8.60 ± 0.01 ^h^	30.78 ± 0.01 ^g^	8.08	55.75
	0.75%	74.30 ± 0.07 ^m^	−9.30 ± 0.03 ^l^	32.87 ± 0.04 ^l^	13.37	57.25
	1.00%	74.54 ± 0.06 ^n^	−9.55 ± 0.02 ^m^	33.08 ± 0.02 ^m^	13.69	57.18

* Mean (±standard deviation) values with different letters within the same column indicate significant difference at *p* < 0.05; *n* = 6. ** ACE, calcium acetate; CAR, calcium carbonate; CIT, calcium citrate; LAC, calcium lactate.

**Table 5 foods-12-02696-t005:** Sensory evaluation analysis * of the cooked noodles at different concentrations of calcium citrate fortification.

Sample	Cooked Noodle **					
	Color	Tissue	Adhesiveness	Springiness	Flavor	Overall Acceptability
Control	4.45 ± 1.02 ^a^	4.75 ± 1.03 ^b^	4.57 ± 1.12 ^b^	4.83 ± 1.21 ^b^	4.51 ± 0.94 ^b^	4.92 ± 1.02 ^b^
0.25%	4.22 ± 1.07 ^a^	4.54 ± 1.16 ^ab^	4.12 ± 1.23 ^a^	4.29 ± 1.33 ^a^	4.15 ± 1.29 ^ab^	4.23 ± 1.14 ^a^
0.50%	4.58 ± 1.01 ^a^	4.34 ± 1.12 ^ab^	4.28 ± 1.05 ^ab^	4.34 ± 1.31 ^a^	4.25 ± 1.06 ^b^	4.34 ± 1.03 ^a^
0.75%	4.51 ± 0.97 ^a^	4.29 ± 1.35 ^ab^	3.95 ± 1.23 ^a^	4.15 ± 1.39 ^a^	4.17 ± 1.15 ^ab^	4.14 ± 1.14 ^a^
1.00%	4.55 ± 1.09 ^a^	4.12 ± 1.27 ^a^	3.88 ± 0.96 ^a^	4.11 ± 1.29 ^a^	3.80 ± 1.02 ^a^	4.08 ± 1.25 ^a^

* 1–7 scale: 1 = extremely dislike, 7 = extremely like. ** Mean of 65 observations. Means within the same column with different letters indicate significant different (*p* < 0.05); *n* = 3.

## Data Availability

The data presented in this work are available on request from the corresponding author.

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
