# Peer review of "Comparison of Physicochemical Properties of Noodles Fortified with Commercial Calcium Salts versus Calcium Citrate from Oyster Shells"

_foods, 2023, doi:10.3390/foods12142696_

Round 1

Reviewer 1 Report

1.What is the purity of citric acid prepared from oyster shells?

2.The water content of noodles with different calcium content is different. Is the difference of physical and chemical properties affected by calcium or water content?

3.What is the maximum amount of additive allowed in noodles?

4. The calcium citrate we produced from oyster shells, due to its smaller particle size, is likely to have higher bioavailability than the calcium citrate salt sample we purchased. Is there any basis for this conclusion?

  •  

Author Response

x

Responses to Comments and Suggestions for Authors

Foods

Title: Comparison of Physicochemical Properties of Noodles Fortified with Commercial Calcium Salts versus Calcium Citrate from Oyster Shells

Dear Reviewer #1

The authors are extremely grateful to anonymous reviewer involved for providing his/her excellent comments and valuable advice in this paper. We have revised the paper based on the reviewer’s comments. We have pleasure in requesting the reviewer to review this paper. Thank you. Your prompt attention to this paper is very much appreciated.

Comments and Suggestions for Authors

Point 1: What is the purity of citric acid prepared from oyster shells?

Response 1: 

The purity of citric acid which we purchased from J.T. Baker (Phillipsburg, NJ, USA) was over 99.98%. And the Infrared spectra of oyster shells is as following Figure 1. Thanks for the question and we very much appreciate your consideration on this matter.

Figure 1 IR spectra of oyster shell powders.

Point 2: The water content of noodles with different calcium content is different. Is the difference of physical and chemical properties affected by calcium or water content?

Response 2: Calcium salts did not change in water activities in raw noodle but in dried noodles. For example, calcium citrate decreases the water content of raw and dried noodles, but it significantly increased the breaking force. Therefore, both water and calcium content influence physical and chemical properties of raw, dried and cooked noodles in pH, cooking losses, cooking yield, breaking force and L* values of dried noodle, and tensile force and elongation of raw noodles.

Point 3: What is the maximum amount of additive allowed in noodles?

Response 3: Our data show the control, 0.25%, 0.50%, 0.75%, and 1.99% calcium salt fortified dried noodle containing 22 ppm, 218-239 ppm, 407-449 ppm, 611-638 ppm, and 792-891 ppm calcium as attached Figure 2. The maximum level of calcium daily dietary intake is not over 1500 mg calcium for children and adults. The recommended dietary allowances (RDA) for calcium is 600 mg for adults and 1-9 year children (Srinivasan and Harinarayan, 2015). Therefore, we are not recommending to eat the 0.50% calcium citrate fortified dried noodles over 6 kg. Thanks for the question and we very much appreciate your consideration on this matter again.

 Figure 2 Effect of kinds and amounts of calcium salts on the calcium content of drying noodles.

ACE, Calcium acetate; CAR, Calcium carbonate; CIT, Calcium citrate; LAC, Calcium lactate.

Point 4: The calcium citrate we produced from oyster shells, due to its smaller particle size, is likely to have higher bioavailability than the calcium citrate salt sample we purchased. Is there any basis for this conclusion?

Response 4: Heller et al. (1999) reported the bioavailability of calcium citrate was higher than that of calcium carbonate. However, the small particle size of calcium citrate made from oyster shells may be much easier dissolve into water and convert to calcium ion, but it may not have higher bioavailability than those of commercial available calcium citrate salt.

Yours truly,

Wen-Chieh Sung, Ph.D.

Professor

Department of Food Science

National Taiwan Ocean University

Reviewer 2 Report

The manuscript entitled "Comparison of Physicochemical Properties of Noodles Fortified with Commercial Calcium Salts versus calcium citrate from Oyster Shells" presents an innovative and interesting study. However, there are a few flaws that need to be addressed before publication. Here are some specific comments for improvement:

Abstract:

·         Lack of clarity regarding the physicochemical effects: The abstract mentions examining the physicochemical effects of fortifying noodles with calcium salts, but it does not provide specific details about the observed effects. It would be beneficial to mention key parameters or properties that were analyzed and affected by the fortification.

·         Incomplete information on the impact of calcium citrate fortification: The abstract states that fortification with calcium citrate had no significant influence on overall scores for cooked noodles, but it does not provide further details on the specific attributes or characteristics that were evaluated in these scores. Elaborating on the sensory attributes assessed would enhance the understanding of the results.

·         Limited comparison between calcium citrate sources: The abstract mentions comparing noodles fortified with calcium citrate made from oyster shells to a control sample and noodles fortified with commercially available calcium citrate. However, it does not provide any specific findings or comparisons between these different sources of calcium citrate. Adding a brief summary of the observed differences or similarities would strengthen the abstract.

·         Lack of discussion on the significance of particle size difference: The abstract briefly mentions a difference in particle size between calcium citrate made from oyster shells and the purchased calcium citrate. However, it does not discuss the potential implications or relevance of this difference in relation to the fortification or noodle quality. Providing some insight or speculation on the significance of particle size variation would be helpful.

·         Absence of broader implications and applications: The abstract does not explicitly mention the broader implications or potential applications of the findings. Including a sentence or two on how the results could be relevant to the food industry, nutrition, or consumer health would enhance the abstract's impact.

Introduction:

·         Lack of a clear research objective: The introduction does not explicitly state the research objective or hypothesis of the study. It would be beneficial to provide a concise statement that outlines the purpose of the research, such as investigating the effects of calcium fortification on white salted noodles.

·         Insufficient background information on calcium fortification in noodles: While the introduction mentions the popularity of fortified noodles in Taiwan and the development of calcium supplements from oyster shells and limestone, it does not provide sufficient context on the current state of calcium fortification in noodles or the specific challenges and research gaps in this area. Providing a brief overview of existing research or the lack thereof would help set the stage for the study.

·         Lack of a clear transition between paragraphs: The introduction jumps between different topics without clear transitions. It would be helpful to structure the paragraphs in a more cohesive manner, ensuring a smooth flow of information and logical progression of ideas.

·         Limited discussion of the importance of calcium and its sources: Although the introduction briefly mentions the nutritional value of calcium and its role in bone health, it does not sufficiently emphasize the significance of calcium fortification in addressing potential dietary deficiencies or the potential benefits of using calcium from oyster shells. Providing more context and discussing the potential impact of calcium fortification on public health or addressing specific nutritional concerns would enhance the introduction.

Addressing these weaknesses would improve the introduction by clearly stating the research objective, providing a comprehensive background on calcium fortification in noodles, improving the flow between paragraphs, and emphasizing the importance and potential benefits of calcium fortification from oyster shells.

Material and Methods:

·         It is essential to clearly emphasize the number of repetitions of measurements throughout the preparation and processing of materials and samples.

·         Move the sentence regarding the use of the dynamic light scattering analyzer (System 4700c sub-micron particle analyzer, Malvern Instruments, Worcestershire, UK) to a more appropriate location in the Material and Methods chapter. It is unclear what specific purpose the device serves, so this should be highlighted.

·         Cite a reference that justifies the procedure for the preparation of calcium citrate from oyster shells (Section 2.2).

Noodle preparation and physicochemical properties of noodles (Section 2.4):

·         Provide a more detailed description of the drying process. Although it was mentioned that drying was carried out at 45°C for 5 hours, it would be beneficial to explain how the decision was made to end the drying after 5 hours and why the specific temperature of 45°C was chosen.

Standard methods and references:

·         When specifying and citing standard methods, such as the example given, "Moisture and ash contents were measured according to the AOAC method [14]," ensure that the number of the method is clearly stated. The same applies to citing literature references in the References chapter.

·         Similar clarification is required for the method mentioned in passing: "Cooking yield and loss were determined according to the AACC method [15]."

·         Specify the reference used for determining Water activity.

Determination of calcium content, color, and texture of noodles (Section 2.5):

·         Clearly emphasize whether the measurements were performed on fresh or dried noodles.

·         Provide an explanation for why the BI index was used instead of hue angle, Chroma value, and Saturation. Reference the relevant literature in this chapter, particularly for calculating the total color change, BI, and similar parameters.

Results:

·         Improve the resolution of Image 1 to enhance its clarity.

Interpretation and references:

·         Provide a reference to support the claim regarding the interpretation of color change related to the browning reaction during drying at 45°C.

Inconsistency regarding WI and BI:

·         In the previously described Material and Methods chapter, BI is mentioned, but there is no mention of WI. Clarify the use and meaning of WI in the tables, as it was not explained earlier.

Potential Weaknesses in the Conclusion:

·         The conclusion does not provide a summary or analysis of the study's limitations or potential areas for future research.

·         The generalizability of the results to a broader context or population is not addressed.

·         The conclusion lacks a clear statement on the overall implications and practical applications of the study's findings.

The quality of English language in the article "Comparison of Physicochemical Properties of Noodles Fortified with Commercial Calcium Salts versus Calcium Citrate from Oyster Shells" appears to be generally good. The article is well-structured with clear headings and subheadings, making it easy to follow. The sentences are coherent and convey the intended meaning. However, there are a few areas where improvements can be made:

·         Formatting and punctuation: There are inconsistencies in punctuation, such as missing spaces after punctuation marks and inconsistent use of capitalization in headings. These issues should be addressed for better readability.

·         Sentence structure: Some sentences are long and complex, which can make them difficult to understand. Breaking them down into shorter sentences or using clearer phrasing would improve readability.

·         Typographical errors: There are a few typographical errors, such as missing hyphens in compound words ("physiochemical" instead of "physicochemical") and misspelled words ("forti-fied" instead of "fortified"). These errors should be corrected for accuracy.

·         Clarity of expression: In some sections, the wording could be improved for clarity and precision. For example, the phrase "Noodles fortified with calcium citrate made from oyster shells were not significantly different from noodles fortified with the pur-chased calcium citrate" could be rephrased as "Noodles fortified with calcium citrate made from oyster shells showed no significant difference compared to noodles fortified with commercially available calcium citrate."

Overall, the article provides clear information about the study and its findings, but attention to the aforementioned language issues would enhance its quality.

Author Response

Responses to Comments and Suggestions for Authors

Foods

Title: Comparison of Physicochemical Properties of Noodles Fortified with Commercial Calcium Salts versus Calcium Citrate from Oyster Shells

Dear Reviewer #2

The authors are extremely grateful to anonymous reviewer involved for providing his/her excellent comments and valuable advice in this paper. We have revised the paper based on the reviewer’s comments. We have pleasure in requesting the reviewer to review this paper. Thank you. Your prompt attention to this paper is very much appreciated.

Comments and Suggestions for Authors

The manuscript entitled "Comparison of Physicochemical Properties of Noodles Fortified with Commercial Calcium Salts versus calcium citrate from Oyster Shells" presents an innovative and interesting study. However, there are a few flaws that need to be addressed before publication. Here are some specific comments for improvement:

Abstract:

  • Lack of clarity regarding the physicochemical effects: The abstract mentions examining the physicochemical effects of fortifying noodles with calcium salts, but it does not provide specific details about the observed effects. It would be beneficial to mention key parameters or properties that were analyzed and affected by the fortification.
  • Incomplete information on the impact of calcium citrate fortification: The abstract states that fortification with calcium citrate had no significant influence on overall scores for cooked noodles, but it does not provide further details on the specific attributes or characteristics that were evaluated in these scores. Elaborating on the sensory attributes assessed would enhance the understanding of the results.
  • Limited comparison between calcium citrate sources: The abstract mentions comparing noodles fortified with calcium citrate made from oyster shells to a control sample and noodles fortified with commercially available calcium citrate. However, it does not provide any specific findings or comparisons between these different sources of calcium citrate. Adding a brief summary of the observed differences or similarities would strengthen the abstract.
  • Lack of discussion on the significance of particle size difference: The abstract briefly mentions a difference in particle size between calcium citrate made from oyster shells and the purchased calcium citrate. However, it does not discuss the potential implications or relevance of this difference in relation to the fortification or noodle quality. Providing some insight or speculation on the significance of particle size variation would be helpful.
  • Absence of broader implications and applications: The abstract does not explicitly mention the broader implications or potential applications of the findings. Including a sentence or two on how the results could be relevant to the food industry, nutrition, or consumer health would enhance the abstract's impact.

Response for comments of Abstract: 

We have revised abstract, rewritten and rechecked the manuscript in our article carefully in the revised abstract, as shown in red, to improve the clarity. And we added more information on the impact of calcium citrate fortification for sensory score of cooked noodles. Also, we have revised the last sentence of abstract to give a broader implications and applications for noodle manufacture.

        However, the small particle size of calcium citrate made from oyster shells may be much easier dissolve into water and convert to calcium ion, but it may not have higher bioavailability than those of commercial available calcium citrate salt. We add more discussions in the section of Results and Discussion.

Point 2: Introduction:

  • Lack of a clear research objective: The introduction does not explicitly state the research objective or hypothesis of the study. It would be beneficial to provide a concise statement that outlines the purpose of the research, such as investigating the effects of calcium fortification on white salted noodles.
  • Insufficient background information on calcium fortification in noodles: While the introduction mentions the popularity of fortified noodles in Taiwan and the development of calcium supplements from oyster shells and limestone, it does not provide sufficient context on the current state of calcium fortification in noodles or the specific challenges and research gaps in this area. Providing a brief overview of existing research or the lack thereof would help set the stage for the study.
  • Lack of a clear transition between paragraphs: The introduction jumps between different topics without clear transitions. It would be helpful to structure the paragraphs in a more cohesive manner, ensuring a smooth flow of information and logical progression of ideas.
  • Limited discussion of the importance of calcium and its sources: Although the introduction briefly mentions the nutritional value of calcium and its role in bone health, it does not sufficiently emphasize the significance of calcium fortification in addressing potential dietary deficiencies or the potential benefits of using calcium from oyster shells. Providing more context and discussing the potential impact of calcium fortification on public health or addressing specific nutritional concerns would enhance the introduction.

Addressing these weaknesses would improve the introduction by clearly stating the research objective, providing a comprehensive background on calcium fortification in noodles, improving the flow between paragraphs, and emphasizing the importance and potential benefits of calcium fortification from oyster shells.

Response for comments of Introduction:

We have added some references to enhance the background information on calcium fortification in noodles and importance of calcium, as shown in red, and the last paragraph of Introduction has been revised to give a clear research objective.

        Thanks for the suggestions and we very much appreciate your consideration on this matter. Hopefully, it has a great improvement on the section.

Point 3: Material and Methods:

  • It is essential to clearly emphasize the number of repetitions of measurements throughout the preparation and processing of materials and samples.
  • Move the sentence regarding the use of the dynamic light scattering analyzer (System 4700c sub-micron particle analyzer, Malvern Instruments, Worcestershire, UK) to a more appropriate location in the Material and Methods chapter. It is unclear what specific purpose the device serves, so this should be highlighted.
  • Cite a reference that justifies the procedure for the preparation of calcium citrate from oyster shells (Section 2.2).

Noodle preparation and physicochemical properties of noodles (Section 2.4):

  • Provide a more detailed description of the drying process. Although it was mentioned that drying was carried out at 45°C for 5 hours, it would be beneficial to explain how the decision was made to end the drying after 5 hours and why the specific temperature of 45°C was chosen.

Standard methods and references:

  • When specifying and citing standard methods, such as the example given, "Moisture and ash contents were measured according to the AOAC method [14]," ensure that the number of the method is clearly stated. The same applies to citing literature references in the References chapter.
  • Similar clarification is required for the method mentioned in passing: "Cooking yield and loss were determined according to the AACC method [15]."
  • Specify the reference used for determining Water activity.

Determination of calcium content, color, and texture of noodles (Section 2.5):

  • Clearly emphasize whether the measurements were performed on fresh or dried noodles.
  • Provide an explanation for why the BI index was used instead of hue angle, Chroma value, and Saturation. Reference the relevant literature in this chapter, particularly for calculating the total color change, BI, and similar parameters.

Response for comments of Materials and Methods:

  • We have corrected the sentences regarding the use of the dynamic light scattering analyzer to section 2.3 (in red) at page 3.
  • We added 9 references to give more detailed information for the procedure for the preparation of calcium citrate from oyster shells, noodle preparation, tests for physicochemical properties of noodles (moisture, ash, cooking yield and loss, water activity, calcium content, color, texture), detailed description of noodle drying process.
  • The mistake has been revised in section 2.5 as red marked texts. And we also added the information for measuring the breaking force of dried noodles in this section. Thank you for pointing out these problems.
  • We added the statement “All analysis was carried out in three replications per treatment unless described otherwise” to emphasize the number of repetitions of measurements in the first red marked sentence at the section of 2.7 Statistical analysis.

Point 4: Results:

  • Improve the resolution of Image 1 to enhance its clarity.

Interpretation and references:

  • Provide a reference to support the claim regarding the interpretation of color change related to the browning reaction during drying at 45°C.

Inconsistency regarding WI and BI:

  • In the previously described Material and Methods chapter, BI is mentioned, but there is no mention of WI. Clarify the use and meaning of WI in the tables, as it was not explained earlier.

Response comments for Results:

Image of Figure 1 (A) pink color infrared spectra curve have been revised to black color curve to enhance its clarity. The sentence refers to the wrong claim regarding the interpretation of color change related to the browning reaction during drying at 45°C has been deleted. We added the sentence “The noodle manufacturers generally use the temperature lower than 45℃ to operate drying process (Zang et al., 2020)” at section 2.4 of Materials and Methods to explain why we used this drying temperature.

Point 5: Potential Weaknesses in the Conclusion:

  • The conclusion does not provide a summary or analysis of the study's limitations or potential areas for future research.
  • The generalizability of the results to a broader context or population is not addressed.
  • The conclusion lacks a clear statement on the overall implications and practical applications of the study's findings.

Response comments for the Conclusions:

We have revised the Conclusions section and provided a summary of the study’s potential areas for future research. The generalizability of the results to a broader utilization of oyster shell resources and current calcium deficiency population for practical applications based on our results in the conclusion of the revised manuscript.

Point 6: Comments on the Quality of English Language

The quality of English language in the article "Comparison of Physicochemical Properties of Noodles Fortified with Commercial Calcium Salts versus Calcium Citrate from Oyster Shells" appears to be generally good. The article is well-structured with clear headings and subheadings, making it easy to follow. The sentences are coherent and convey the intended meaning. However, there are a few areas where improvements can be made:

  • Formatting and punctuation: There are inconsistencies in punctuation, such as missing spaces after punctuation marks and inconsistent use of capitalization in headings. These issues should be addressed for better readability.
  • Sentence structure: Some sentences are long and complex, which can make them difficult to understand. Breaking them down into shorter sentences or using clearer phrasing would improve readability.
  • Typographical errors: There are a few typographical errors, such as missing hyphens in compound words ("physiochemical" instead of "physicochemical") and misspelled words ("forti-fied" instead of "fortified"). These errors should be corrected for accuracy.
  • Clarity of expression: In some sections, the wording could be improved for clarity and precision. For example, the phrase "Noodles fortified with calcium citrate made from oyster shells were not significantly different from noodles fortified with the pur-chased calcium citrate" could be rephrased as "Noodles fortified with calcium citrate made from oyster shells showed no significant difference compared to noodles fortified with commercially available calcium citrate."

Overall, the article provides clear information about the study and its findings, but attention to the aforementioned language issues would enhance its quality.

Response comments on the Quality of English Language:

  • We have reviewed the manuscript thoroughly make corrections for grammar mistakes, typos, punctuations, and format in the manuscript. Additionally, we have asked we have asked Wallace academic editing for English proofreading, and the certificate of editing for has been attached. We hope these could help improve the clarity of this manuscript.
  • The phrase "Noodles fortified with calcium citrate made from oyster shells were not significantly different from noodles fortified with the purchased calcium citrate" in abstract was rephrased as "Noodles fortified with calcium citrate made from oyster shells showed no significant difference compared to noodles fortified with commercially available calcium citrate”.

Thank you for your suggestions and we very much appreciate your consideration on this matter.

Yours truly,

Wen-Chieh Sung, Ph.D.

Professor

Department of Food Science

National Taiwan Ocean University